# Research on the Coupled Risk of Key Nodes in Maritime Transport Based on Improved Catastrophe Theory

**Baode Li, Jing Li * and Jing Lu**

College of Transportation Engineering, Dalian Maritime University, Dalian 116026, China
* Correspondence: lijing@dlmu.edu.cn; Tel.: +86-0411-8472-4241

**Abstract:** Defining the degree of risk of maritime transport nodes is an important basis for studying the security status of maritime transport passages. However, some important straits or canals are key nodes in the maritime transport passage, and their system security conditions are affected by the interaction of uncertainty risk factors. This study addresses the issue of the security status of transport nodes from the perspective of the characteristics of influencing factors. With a focus on catastrophes and the mutual coupling characteristics of the factors that influence the security of maritime transport nodes, a model based on principal component analysis (PCA) and catastrophe theory (CT) is constructed, and the coupled risk degree of the key nodes in Chinese maritime transportation is empirically analysed. The results show that the Taiwan Strait has the lowest risk degree and that the Windward Strait has the highest risk degree among the key nodes in Chinese maritime transport. From the perspective of the security status of these key nodes, most nodes are currently in a stable and safe state.

**Keywords:** waterway transportation; key nodes; coupled risk; PCA; catastrophe theory

## 1. Introduction

As a major part of the maritime transport passage, the security of maritime transport node has been the focus of research in recent years. Due to the fact that the security situation is affected by factors from the internal and external environment, the research on the security of the single maritime transport node and the mechanism of the influencing factors are gradually increasing. In addition, the research methods used are also different, among which the catastrophe theory is the representative one. At present, relevant researches around the above are as follows:

### 1.1. Key Nodes

A maritime transport node is a specific maritime area with a special geographical location or special supply function for ships at sea; such nodes include canals, straits, and important ports [1]. With the emergence of nontraditional security issues at sea, research on the security of key nodes in maritime transport has received increasing attention. Notably, security studies have been conducted on specific key nodes, such as the Strait of Malacca; specifically, some scholars have discussed its security [2], and some have proposed security measures related to the impacts of specific risk factors [2,3]. In addition, security analyses of some other key transportation nodes, such as the Panama Canal, the Suez Canal, the Strait of Hormuz, and the Turkish Straits, have been conducted from different perspectives [4–7]. There are also studies on maritime transport based on the strait or canal. Gao, T. and Lu, J. [8] used a mathematical model based on a programming formulation to measure the impact of a blockage of a strait or canal on the Chinese fleet. Wu Di et al. [9] analysed statistical data on all routes operated by

the top 100 global container liner companies and constructed a network model to study the impact of main channel interruption on container shipping. Zhi, H. et al. [10] discussed the proposed Kra Canal and its impacts on the tanker market from an economic perspective.

At present, it is mainly to study the specific aspects of a node or to study the maritime based on nodes. However, security analyses based on the interaction between influencing factors are lacking. Therefore, this paper takes the main straits and canals as key nodes and takes these node systems as research objects, and considers the relationship of factors to conduct security analysis.

## 1.2. Risk Coupling of Key Nodes

The term "coupling" was originally derived from physics, which refers to the phenomenon of joint interaction between two or more systems or forms of motion through various forms of interaction [11]. Coupling can have different definitions in different areas. In the field of risk management, risk coupling refers to the occurrence of certain types of risks in the system and the extent to which its influence depends on other risks, and the extent to which the risks affect other risk formations and influences [12]. In terms of maritime transport passages, there is a coupling effect between the various risk factors affecting the maritime transport passage; that is, the occurrence of two or more risk factors will produce a certain force or offset effect. If the degree of risk is greater than the simple sum of the risk levels of the two factors, there is a positive coupling effect, and vice versa [13]. Therefore, the degree of risk coupling of maritime transport nodes refers to the safe state formed when these nodes are affected by the mutual interactions of dynamic factors such as bad weather, pirates, maritime terrorism, etc. Research on the coupling mechanisms associated with the risk factors of shipping has gradually increased. For example, the coupling mechanisms of ship navigation risk factors have been studied [14], from the perspective of a complex port–ship navigation system, and a system dynamics method was used to simulate the port–ship navigation risk [15]. In addition, some scholars have conducted security analyses based on the relevant risk coupling mechanisms. Shan, S.F. [16] identified potential risk influencing factors (RIFs) for the Arctic maritime transportation system from human, technical (ship), environmental, and organizational aspects. An analytical hierarchy process method is used to analyse the hierarchical relationships and to calculate the relative importance of the selected RIFs. Gudela Grote [17] used the theory of loose coupling to analyse the security management of high-risk systems and applied the analysis results to optimize human decision-making and improve the security of the system. Peter Brooker [18] studied the question: "what do design security targets really mean and imply for risk modelling?" by analysing the nature of accidents considering the corresponding causal factors and performing practical collision risk modelling.

Currently, although there are more and more studies on risk coupling in maritime transport; however, research on risk coupled of key nodes is still in its infancy, focusing on the identification of risk factors, with a lack of research on the mechanism of action between influencing factors. This paper focuses on risk analysis of important maritime transport nodes from the coupling mechanism of factors affecting maritime transport security.

## 1.3. Application of Catastrophe Theory in Maritime Transport Security

In previous studies, the catastrophe theory was widely used in security evaluations [19]. Nashwan, M.S. [20] used fuzzy catastrophe theory and data clustering methods to solve certain data subjectivity problems. Ahammed, S.J. [21] used a multicriteria decision-making approach based on catastrophe theory to assess systemic indices. In the application of catastrophe theory in maritime security research, in addition to directly using the catastrophe progression method based on the catastrophe theory [22], a fuzzy catastrophe evaluation model was established by combining fuzzy and catastrophe theory to comprehensively evaluate the maritime traffic risk [23]. In other areas of application, Papacharalampous, A.E. [24] proposed a framework based on stochastic cusp catastrophe theory to model microscopic freeway traffic flow. Ahmed, K. [25] used catastrophe theory for the assessment of groundwater potential zones in the arid region of the lower Balochistan province of Pakistan.

Wen, Y.L. [26] used the catastrophe progression method and fuzzy comprehensive evaluation to comprehensively evaluate and predict the security of ship navigation in coastal areas.

Although the catastrophe theory is widely used, it mainly uses the catastrophe progression method based on catastrophe theory. When using the catastrophe progression method, the first step is to establish a hierarchical structure. At present, it is mainly qualitative to establish, thus, it lacks objectivity. This paper has improved upon this, based on the relationship between factors, to quantitatively establish the hierarchical structure.

In view of the above literature, this paper studies the coupled risk of key nodes in maritime transport, based on the mechanism of action between risk factors. Among them, the key nodes of transportation refer to important straits or canals. According to the interaction between risk factors, the traditional catastrophe model is improved to construct a coupled risk measurement model of maritime transport nodes. On the one hand, it can ably analyse the impact of risk factors on key nodes; on the other hand, it can improve the applicability of the catastrophe model so that it can thoroughly analyse the risk coupling status of key nodes in maritime transport. Considering the straits and canals that the important material transportation routes pass through, the key nodes of maritime transportation are selected to conduct empirical research. The key nodes constitute an important foundation and support for the maritime transportation network. Therefore, it is of great significance to analyse the risk degree of the key nodes in maritime transportation networks to aid maritime transportation.

## 2. Methods

### 2.1. Basic Theory

#### 2.1.1. Characteristics of Security Threat Factors in the Nodes of Maritime Transport

The security of maritime transport nodes is affected by various uncertainties. However, by taking various security measures, the transport nodes are generally in a state of stable equilibrium. At a certain point in time, when a new security threat emerges or the degree of influence of the original security threat changes, the original equilibrium state of the node may be disrupted, and the security state of the transport node can shift to "catastrophe." Therefore, the maritime transport node experiences a security threat and resists the external security threat to maintain its original equilibrium state, which comprehensively reflects the security status of the maritime transport node. The risk factors of maritime transport nodes can be characterised as follows.

1.  The degree of security of maritime transport nodes is affected by the mutual coupling of two types of control variables: external disaster factors and various measures to address disasters.
2.  The steady state of maritime transport nodes can suddenly change under different degrees of influence from the above two types of control variables. By measuring the catastrophe level associated with the security of maritime transport nodes, the degree of coupled risk of maritime transport nodes can be effectively evaluated.
3.  There are two stable states before and after the catastrophe state of maritime transport nodes.

#### 2.1.2. Catastrophe Theory

Catastrophe theory was proposed by the French mathematician Rene Thom based on mathematical foundations such as topology and structural stability to address problems involving discontinuous changes and catastrophe [27]. Using different elementary functions to correspond to different types of mutations, theoretical analysis of the system can be applied not only to the natural sciences such as mathematics, physics, and mechanics [28], but also to the fields of social sciences and engineering sciences [29]. Generally, there are multiple stable state systems in the evolution process, which can be analysed by using the catastrophe theory [30].

The catastrophe theory in general refers to the elementary catastrophe theory. The catastrophe model based on catastrophe theory is described by a potential function, which is determined by state

and control variables, where the state variables represent the behavioural state of the system and the control variables represent various factors that influence the state variables. Let $f(x,c)$ denote the system potential function, where $x$ is the state variable and $c$ is the control variable. Then, the critical point equation can be expressed as follows.

$$\frac{df(x,c)}{dx} = 0 \tag{1}$$

Equation (1) is a balanced system composed of all critical points. The system is unstable near the critical point of degradation. If the control variables slightly change, it may cause a catastrophe in the system. The critical point of degradation satisfies the following conditions.

$$\begin{cases} f'(x,c) = 0 \\ f''(x,c) = 0 \end{cases} \tag{2}$$

The bifurcation point set equation is obtained by eliminating $x$ in (2), and the bifurcation point set equation encompasses the point of change of all forms of the potential function. The normalization formula can be derived using the bifurcation point set equation. The normalization formula normalizes the different states of the control variables of the system into the same mass state and uses it to perform recursive operations. The value of the total catastrophe membership function representing the state of the system is obtained as a basis for the comprehensive measurement.

Rene Thom demonstrates that the number of discontinuous constructs that may differ in nature is determined primarily by the number of control variables, not by the number of state variables. According to the proof of Rene Thom, a large amount of discontinuous catastrophe in both natural and social sciences can be described by corresponding geometric shapes, and if there are no more than four elements in the control variable set $c$, then the system potential function has at most 7 catastrophe models: fold, cusp, swallowtail, butterfly, hyperbolic, elliptical, and parabolic; wherein the state variable is a one-dimensional potential function and the corresponding normalization formula is shown in Table 1.

**Table 1.** One-dimensional catastrophe models of the state variable.

| Model Name | Control Variable | Potential Function | Normalization Formula |
|---|---|---|---|
| Fold | $1(u)$ | $f(x) = x^3 + ux$ | $x_u = u^{1/2}$ |
| Cusp | $2(u,v)$ | $f(x) = x^4 + ux^2 + vx$ | $x_u = u^{1/2}, x_v = v^{1/3}$ |
| Swallowtail | $3(u,v,w)$ | $f(x) = x^5 + ux^3 + vx^2 + wx$ | $x_u = u^{1/2}, x_v = v^{1/3}, x_w = w^{1/4}$ |
| Butterfly | $4(t,u,v,w)$ | $f(x) = x^6 + tx^4 + ux^3 + vx^2 + wx$ | $x_t = t^{1/2}, x_u = u^{1/3}, x_v = v^{1/4}, x_w = w^{1/5}$ |

Note: $x_t$, $x_u$, $x_v$ and $x_w$ are the catastrophe progression values corresponding to each control variable.

When using the catastrophe theory to measure, the catastrophe progression method based on the catastrophe theory is used. Applying the catastrophe progression method, we must first perform multilevel contradiction decomposition on the evaluation target of the system, and then use the catastrophe fuzzy membership function generated by the combination of catastrophe theory and fuzzy mathematics to obtain the elementary catastrophe model and its bifurcation point set equation. The two can derive the normalization formula, and the generalized formula is used for the comprehensive quantization operation. Finally, the normalization is a parameter; that is, the total membership function is obtained, and the comprehensive evaluation is performed.

### 2.1.3. Principal Component Analysis

Principal component analysis (PCA) uses the ideas of dimension reduction to convert multiple variables to a few variables, in which each principal component is a linear combination of the original variables, and each principal component is unrelated. These principal components can reflect most of the information of the original variables and contain information overlapping each other [31].

The PCA model can be briefly described as follows.

Suppose $p$ variables are used to describe the research objects, respectively using $X_1, X_2, \cdots, X_p$ as representatives. The $p$-dimensional random vector formed by these $p$ variables is $X = (X_1, X_2, \cdots, X_p)^T$. Let $a$ denote the mean of random variables, and the linear combination of the original variables is,

$$
\begin{cases}
F_1 = a_{11}X_1 + a_{21}X_2 + \cdots + a_{p1}X_p \\
F_2 = a_{12}X_1 + a_{22}X_2 + \cdots + a_{p2}X_p \\
\quad \cdots \\
F_p = a_{1p}X_1 + a_{2p}X_2 + \cdots + a_{pp}X_p
\end{cases} \tag{3}
$$

The principal component is an uncorrelated linear combination $F_1, F_2, \cdots, F_p$, and $F_1$ is the largest variance of the linear combination of $F_1, F_2, \cdots, F_p$.

The basic steps of using principal component analysis are as follows.

1. Because the ranges of values and the units of measurement of the original data sets are different, the indices should be normalized.

2. Establish the correlation coefficient matrix $R$ based on standardized data matrix. $R$ reflects the degree of correlation between the standardized data, where $R_{ij}(i, j = 1, 2, \cdots, p)$ is the correlation coefficient between the original variables $X_i$ and $X_j$, and the calculation formula is,

$$
R_{ij} = \frac{\sum\limits_{l=1}^{n} (X_{lj} - X_i)(X_{lj} - X_j)}{\sqrt{\sum\limits_{l=1}^{n} (X_{lj} - X_i)^2 (X_{lj} - X_j)^2}} \tag{4}
$$

3. Calculate the eigenvalue and the cumulative variance contribution rate based on the correlation coefficient matrix $R$, and then determine the number of principal components, wherein the eigenvalue and the variance contribution rate are calculated as (5) and (6).

$$
|\lambda E - R| = 0 \tag{5}
$$

$$
\omega_i = \lambda_j / \sum_{j-1}^{p} \lambda_j \tag{6}
$$

$\omega_i$ represents the variance contribution rate of the principal component $F_i$. In addition, the formula for calculating the cumulative variance contribution rate is,

$$
\sum_{j-1}^{q} \lambda_j / \sum_{j-1}^{p} \lambda_j \tag{7}
$$

According to the principle of selecting the number of principal components, $1, 2, \cdots, q(q \leq p)$ corresponding to the eigenvalue $\lambda_1, \lambda_2, \cdots, \lambda_q$ of greater than 1 and the cumulative variance contribution rate of 80–95% is required, wherein the integer $q$ is the number of the main components.

4. Establish a factor load matrix to explain the principal components, and the factor load matrix reveals the degree of correlation between principal components and indicators.

$$
b_{ij} = a_{ij} \sqrt{\lambda_i} \tag{8}
$$

where $b_{ij}$ denotes the factor load of the $i$ th index on the $j$ th principal component and $a_{ij}$ denotes the $j$ th component of the feature vector corresponding to the $i$ th eigenvalue.

5. Based on the above results, the principal components are expressed and used for calculation and analysis

In this paper, the PCA method is used to improve the measurement steps of the catastrophe theory, mainly using the first 4 steps of the PCA method, which are specifically referred to in the next section.

*2.2. Improved Catastrophe Theory Model Based on PCA*

2.2.1. Establish Model

According to the characteristics of the security factors affecting the maritime transport node and the fields and scenarios applicable to the catastrophe theory, the catastrophe model can effectively describe the changes in maritime transport nodes from stable equilibrium to an unsteady state. To describe the degree of coupled risk of maritime transport nodes, the concept of the coupling degree is proposed. The degree of coupling is the degree of coupled risk.

Let $G(x, c)$ denote the coupling degree potential function of key nodes, where $x$ is the state variable and $c$ is the control variable. The state variable is the coupling degree of the key nodes, the control variables represent various influential factors, and their joint action determines the coupling degree of the key nodes. The coupling degree and judgement criteria for key nodes can be described as follows.

$$G(x) = f(x, c) \tag{9}$$

$$\frac{dG(x)}{dx} = \frac{\partial f(x, c)}{\partial x} = 0 \tag{10}$$

$$\Delta = \begin{cases} \frac{\partial f(x,c)}{\partial x} = 0 \\ \frac{\partial^2 f(x,c)}{\partial x^2} = 0 \end{cases} \tag{11}$$

Equation (9) represents the degree of coupling of the potential functions of key nodes. Equation (10) is the equilibrium state of the key nodes in the state space, which represents a balance among the risk factors of the key nodes. Equation (11) is a bifurcation point set equation, which can be used to determine whether the key node coupling degree is stable. When $\Delta < 0$, the key node risk coupling degree is unstable; when $\Delta = 0$, the key node risk coupling degree is in a critical stable state, and even a small interference may cause a catastrophe; when $\Delta > 0$, the key node risk coupling degree is stable.

2.2.2. Improved Measurement Model

Most previous studies used qualitative methods to establish a hierarchical model structure. This paper considers the importance of the factors in the established hierarchical structure model. Considering the different degrees of representativeness among the security factors of the key nodes in maritime transportation, some information duplication will occur among the indices. Therefore, to comprehensively measure the degree of coupled risk among the various transport nodes and the objective explanatory factors, this paper uses PCA to establish the hierarchical structure of key nodes, making the approach more objective and representative than traditional methods.

The model improvement steps are as follows.

1. According to the characteristics of the influential factors, these factors can be divided into two categories: external disaster factors, defined as a vulnerability index, and various measures to address disasters, defined as an adaptivity index.
2. PCA was performed on the two types of indices, and the hierarchical structure was established.

(1) Because the ranges of values and the units of measurement of the original data sets are different, the indices should be normalized. This paper adopts standardized formulas such as (12) and (13) for this task.

$$\text{Positive}: \ x_{ij}^{*} = \frac{x_{ij} - \min\limits_{1 \le i \le n}(x_{ij})}{\max\limits_{1 \le i \le n}(x_{ij}) - x_{ij} - \min\limits_{1 \le i \le n}(x_{ij})} \tag{12}$$

$$\text{Negative}: \ x_{ij}^{*} = \frac{\max\limits_{1 \le i \le n}(x_{ij}) - x_{ij}}{\max\limits_{1 \le i \le n}(x_{ij}) - x_{ij} - \min\limits_{1 \le i \le n}(x_{ij})} \tag{13}$$

where $x_{ij}^{*}$ is the value after the influential factor index is processed; $i = (1, 2, \cdots, n)$ denote key nodes; $j = (1, 2, \cdots, m)$ denote the influential factors; $x_{ij}$ is the $j$ th influential factor index value of the $i$ th key node; and $\max(x_{ij})$ and $\min(x_{ij})$ denote the maximum and minimum values of the $j$ th influential factor of the $i$ th key node, respectively.

(2) Calculate the correlation coefficient matrix of the normalized index data $R_{m \times m}$.

(3) Calculate the eigenvalues $\lambda_i (i = 1, 2, \cdots, n)$ of the correlation coefficient matrix $R$, where $\lambda_i$ denotes the total variance of the original index data explained by the principal component $F_i$. The variance contribution of the original index data $\omega_i$ denoted by the principal component $F_i$ can be expressed as follows.

$$\omega_i = \lambda_i / \sum_{i=1}^{n} \lambda_i \tag{14}$$

Equation (14) denotes the ratio of the original information represented by the $i$ th principal component $F_i$ to all the original information. The calculated $\lambda_i$ values are sorted from large to small. When the cumulative variance contribution of the $p$ th eigenvalue $\lambda_p$ is greater than or equal to 85%, the principal components corresponding to the first $p$ eigenvalues are selected.

(4) Calculate the factor load matrix to establish the relationships among indices and importance of each index. Finally, the hierarchical structure of the factor indices is established. Related calculation formula reference (8).

3. When using the catastrophe progression method, the obtained values are often clustered. To make the calculated coupling value more intuitively represent the coupling degree of key nodes and facilitate subsequent operations, this paper improves the initial coupling degree value calculated by the traditional catastrophe progression method according to reference [32].

First, according to the established coupled risk degree catastrophe model of key nodes, the comprehensive coupling degree value $k_i$ is calculated for control variable set $\{0, 0.2, 0.4, 0.6, 0.8, 1\}$, and the six values are used as the scales of the initial comprehensive values. The corresponding interval for different scales is $[k_i, k_{i+1}] (i = 0, 1, \cdots, 4)$.

Second, the calculated coupling degree value $K_j \{K_1, K_2, \cdots, K_n\}$ ($n$ denotes key node) is mapped to the corresponding interval according to the scale interval $[k_i, k_{i+1}] (i = 0, 1, \cdots, 4)$ in which it falls, and the adjusted coupling degree value is $K_j' \{K_1', K_2', \cdots, K_n'\}$.

$$K_j' = 0.2 \left[ \left( \frac{K_j - k_i}{k_{i+1} - k_i} \right) + i \right] \tag{15}$$

4. Based on the above analysis, the final determination of the coupling degree of the key nodes considers both the vulnerability index and the adaptivity index. In addition, according to Table 1, the cusp catastrophe model can be used to describe the state of the security system of the key nodes. The cusp catastrophe state of the key node security system based on the equilibrium state and the control plane can be established as shown in Figure 1.

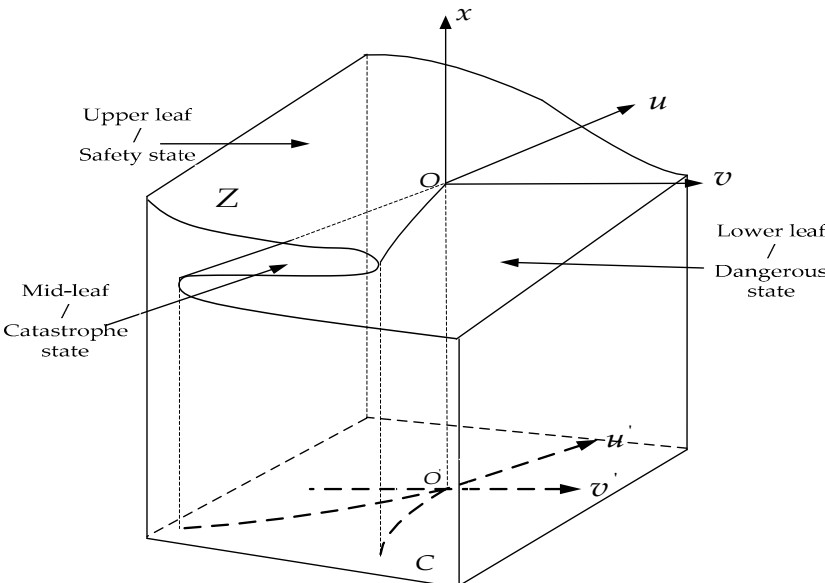

**Figure 1.** Schematic diagram of the cusp catastrophe model of key nodes in marine transportation.

The z-surface in Figure 1 is a balanced surface with a cusp catastrophe, and the c-surface below is the control surface of the control variables $u$ and $v$. The balanced surface consists of the upper, middle, and lower leaves, which correspond to the security state, the catastrophe state, and the dangerous state of the security system of key nodes at sea, respectively. For the control surface, the middle curve is the projection of the surface where the middle leaf is located, which is also the curve formed by the set of bifurcation points. The upper and lower leaves are the stable states of the key node security system, and the middle leaf is in an unstable state that is prone to catastrophe. The security system state of each key node can be assessed by calculating the corresponding Δ value through the bifurcation point set equation. To determine the state of the security system of key nodes based on the values of $u$ and $v$, the bifurcation point set of the cusp mutation formula is transformed as follows.

(1)   If $v > \left[\frac{8}{27}u^3\right]^{1/2}$, the security system of key nodes in maritime transport is in a stable state.

(2)   If $v = \left[\frac{8}{27}u^3\right]^{1/2}$, the security system of key nodes in maritime transport is in a critical steady state.

(3)   If $v < \left[\frac{8}{27}u^3\right]^{1/2}$, the security system of key nodes in maritime transport is in an unstable state.

## 3. Empirical Analysis

This paper employs the key straits and canals in Chinese maritime transport passage as empirical cases to analyze the coupled risk of key nodes in maritime transport [1].

### 3.1. Selection of Key Nodes in Chinese Maritime Transportation

According to statistics from the general administration of customs, P.R. China, China imported 462 million tons of crude oil, 1.064 billion tons of iron ore, 281 million tons of coal and 108 million tons of grain and international container throughput of the above-scale ports was 249.82 million twenty-feet equivalent unit (TEU) in 2018, of which more than 90% were transported by shipping completed. Some straits or canals are the important part of the transport passages of these important materials, and it is also the place where the above materials must pass through. In addition, from the perspective of Chinese important strategic material import passages and routes distribution [33], the main sea import transportation passages for crude oil are (1) Persian Gulf - Hormuz Strait - Malacca Strait - (or Makassar Strait) - Taiwan Strait - China; (2) North Africa-Mediterranean- Gibraltar Strait - Cape of Good Hope -Malacca Strait-Taiwan Strait-China; (3) West Africa - Cape of Good Hope - Malacca Strait -Taiwan Strait-China; the main sea import transportation passages for iron ore are: (1) Australia-Pacific-

China; (2) Brazil - Magellan (Panama) - Pacific - China; (3) Brazil - Cape of Good Hope - Indian Ocean - Malacca Strait (Sunta Strait, Makassar Strait) - Taiwan Strait - China; (4) South Africa - Indian Ocean - Malacca Strait - South China Sea - Taiwan Strait - Mainland China; (5) India - Indian Ocean - Malacca Strait - South China Sea - Taiwan Strait - Northern ports; the main sea import transportation passages for coal are: (1) Australia - Pacific (Ryukyu Islands) - China Mainland; Vietnam - Joan State Strait - Taiwan Strait - Northern ports; (2) Indonesia - Malacca - South China Sea - China; (3) South Africa - Cape of Good Hope - Malacca Strait (Sunta Strait, Makassar Strait) - South China Sea - China; (4) North America - China mainland route. From the above, some important straits and canals can be found to be an important part of the strategic material transportation passage. Therefore, this paper concludes that the straits and canals are the key nodes of Chinese maritime transportation, based on the important straits and canals through which different Chinese important material routes pass and the important roles of the straits or canals, as shown in Table 2.

### 3.2. Construction of a Coupled Risk Measurement System for Key Nodes

#### 3.2.1. Security Factors

The factors that influence the navigation status of key nodes in maritime transport mainly include two categories: traditional and nontraditional threats. Since the end of the Cold War, the threat of traditional military confrontations to maritime transport nodes has decreased, but military conflicts in some areas and the military control of key transport nodes in certain countries still pose important threats to the navigation conditions of maritime transport nodes. In addition, extreme weather events such as typhoons, heavy fog, and ice flows; natural factors such as large waves, earthquakes, and tsunamis; and security threats such as pirates and maritime terrorism have become increasingly serious. The most seriously affected areas include the Gulf of Aden and the Strait of Malacca.

In response to security threats, the countries in which the key nodes are located and the countries in which they are used will also take measures to prevent or mitigate disasters caused by risk factors, such as the introduction of various legal policies, the establishment of specialized guarantee institutions, and the strengthening of cooperation among countries. For example, In response to the Malacca Strait, there are laws or agreements such as the "Regional cooperation agreement on combating piracy and armed robbery of ships in Asia," the "Joint declaration on cooperation in non-traditional security fields between China and ASEAN," and memorandums of understanding to strengthen cooperation against piracy or maritime terrorism.

#### 3.2.2. Risk Coupling Measurement System

According to the characteristics of the influencing factors and reference [1], this paper considers two types of factors that affect the security of key nodes in maritime transport as first-level indices for measuring the security of these nodes. The vulnerability index values of key nodes in maritime transportation reflect the susceptibility of the nodes to external disturbances, mainly based on the stability of the system structure. The adaptivity index of key nodes in maritime transport encompasses the disaster response and response capabilities of the system and the system resilience, i.e., the extent to which the system can avoid damage and rebound after a disaster event. The second-level indices are shown in Table 3. The second-level indices will change with time and the risk degree. The second-level indices associated with vulnerability are negative indices, and those related to adaptivity are positive indices.

**Table 2.** Key nodes of Chinese maritime transportation.

| Strait/Canal | Affiliation Passage | Important Role | Transport Important Material |
|---|---|---|---|
| Taiwan Strait | China's coastal areas - Southeast Asia, China's coastal areas - Africa, China's coastal areas - Europe, China's coastal areas - the Middle East, China's coastal areas - South America's east coast, China's coastal areas - Australia | The main channel between the East China Sea and its northern neighbouring seas and the South China Sea and the Indian Ocean | Crude oil, iron ore, coal, grain, container |
| Malacca Strait | China's coastal areas - Africa, China's coastal areas - Europe, China's coastal areas - North America's east coast - Middle East, China's coastal areas - South America's east coast | Important channel between the Indian Ocean and the Pacific Ocean; Important channel from West Asia to East Asia | Crude oil, iron ore, container |
| Mande Strait | China's coastal areas - North Africa, China's coastal areas - Europe | Located along the shortest route from the Atlantic Ocean to the Indian Ocean and is an important channel for maritime trade between Europe, Asia, and Africa | Crude oil, container |
| Suez Canal | China's coastal areas - North Africa, China's coastal areas - Europe | A famous international waterway connecting the Mediterranean Sea and the Red Sea; essential for maritime navigation in the North Atlantic, Indian Ocean, and Western Pacific | Container |
| Lombok Strait | Indian Ocean - China's coastal areas, China's coastal areas - Australia/New Zealand | Provides a connection to the Indonesian Archipelago; important channel for maritime shipping in the Pacific Ocean and the Indian Ocean | Crude oil, iron ore, grain |
| Sunta Strait | Indian Ocean - China's coastal areas | An important sea route between the Pacific Ocean and the Indian Ocean; on the route from North Pacific countries to East Africa, West Africa; also a detour to the Cape of Good Hope and Europe | Crude oil |
| Makassar Strait | Indian Ocean - China's coastal areas, China's coastal areas - Australia/New Zealand | Along an important route to the South China Sea and from the Philippines to Australia | Iron ore, grain |
| Gibraltar Strait | China's coastal areas - Europe, China's coastal areas - North America's east coast | The only channel from the Mediterranean to the Atlantic Ocean is also the throat of Western European and Northern European countries through the Mediterranean Sea, the Suez Canal, and the Indian Ocean | Crude oil, container |
| Hormuz Strait | China's coastal areas - Middle East | Persian Gulf oil must pass through this sea channel on the way to Western Europe, the United States, Japan and the rest of the world | Crude oil, container |
| Panama Canal | China's coastal areas - North America's east coast | Important channel connecting to the Pacific Ocean and Atlantic Ocean | Crude oil, grain, container |
| Korean Strait | China's coastal areas - Japan | Provides convenient access to the sea between the Japanese Archipelago and the Asian continent; the only channel connecting the Sea of Japan to the East China Sea and the Yellow Sea | Iron ore, grain, container |
| English Channel | China's coastal areas - Western European countries, North America's east coast - Caribbean routes | The channel connects the Atlantic Ocean and the North Sea | Container |

**Table 2.** *Cont.*

| Strait/Canal | Affiliation Passage | Important Role | Transport Important Material |
| --- | --- | --- | --- |
| Florida Strait | Western Europe - North America's south coast | An important channel connecting the Gulf of Mexico and the Atlantic Ocean | Grain, container |
| Dagu Strait | China's coastal areas - North America's east coast, China's coastal areas - North America's west coast | Sea channel between the ports of the East China Sea, the Yellow Sea coast and the east coast ports of Japan | Grain, container |
| Zonggu Strait | China's coastal areas - North America's west coast | An important entrance to the seas of Japan | Container |
| Mindoro Strait | China's coastal areas - Australia/New Zealand | An important channel along the route from China to the Indian Ocean | Iron ore, grain |
| Windward Strait | Northwest Europe/North America's east coast - Caribbean routes | An important channel from the Atlantic Ocean to the Caribbean | Grain, container |
| Mona Strait | Northwest Europe/North America's east coast - Caribbean routes | An important channel connecting the Caribbean Sea and the Atlantic Ocean | Grain, container |

**Table 3.** Risk factor index system for key nodes in maritime transport.

| First-Level Index | Second-Level Index | | Nature of the Index * |
|---|---|---|---|
| | Index | Index Interpretation | |
| Vulnerability | Military conflict | Number of countries at risk of military conflict along the strait or canal/Number of countries in which the strait or canal is located | Negative |
| | Military base | Number of countries with military bases along the strait or canal/Number of countries in which the strait or canal is located | |
| | Pirates | Average number of pirate attacks per year in the strait or canal | |
| | Maritime terrorism | Number of countries in which maritime terrorism has occurred along the strait or canal/Number of countries in which the strait or canal is located | |
| | Accident (caused by extreme weather, etc.) | Average number of accidents per year in the strait or canal | |
| | Coastal political stability | Number of countries in which the strait or canal is classified as dangerous/Number of countries in which the strait or canal is located | |
| Adaptivity | Alternative | Number of alternative channels for the strait or canal | Positive |
| | Related organization | Number of responsible agencies in the relevant country or new agencies established to coordinate the management of the strait or canal | |
| | Security norm | Number of special agencies established by the country/countries in which the channel or canal is located to ensure the smooth flow of traffic through the strait or canal | |
| | Legal policy | Number of relevant laws and policies for the strait or canal | |
| | International cooperation | Number of international cooperation measures for the strait or canal | |

* indicates the impact of the index on the risk of the key nodes; a positive index will reduce the risk, and a negative index will increase the risk.

*3.3. Coupled Risk Measurement for Key Nodes in Maritime Transport*

3.3.1. Data Sources and Normalization

The data in this paper mainly come from the Global Integrated Shipping Information System (GISIS) database, some statistical reports, and websites. Pirates and the number of accidents in each node are sourced from GISIS, while maritime terrorism in each node is sourced from the "Global Terrorism Database." Military conflict and military bases in each node are sourced from "Hull War, Piracy, Terrorism and Related Perils." Coastal political stability in each node is sourced from "FP Marine Risks." In addition, others are mainly collected from the relevant websites of the country in which each node belongs or the countries in which they are used, such as the international maritime bureau, China maritime bureau, Maritime and Port Authority of Singapore, Suez Canal authority, Panama Canal Authority and others. The data selection period is from 2010 to 2018. Formulas (12) and (13) are used to standardize the index data for the key nodes in Chinese maritime transport. The results are shown in Tables 4 and 5.

**Table 4.** Standardized vulnerability index data for the key nodes.

| Index / Key Node | Military Conflict | Military Base | Pirates | Maritime Terrorism | Accident | Coastal Political Stability |
|---|---|---|---|---|---|---|
| Taiwan Strait | 1.00 | 0.75 | 1.00 | 1.00 | 0.40 | 1.00 |
| Malacca Strait | 0.33 | 0.75 | 0.00 | 0.33 | 0.00 | 0.67 |
| Mande Strait | 0.00 | 1.00 | 0.45 | 1.00 | 0.88 | 0.00 |
| Suez Canal | 1.00 | 0.75 | 0.86 | 0.00 | 0.49 | 1.00 |
| Lombok Strait | 0.00 | 0.75 | 0.48 | 0.00 | 0.93 | 0.00 |
| Sunta Strait | 0.00 | 1.00 | 0.48 | 0.00 | 0.84 | 0.00 |
| Makassar Strait | 0.00 | 0.50 | 0.74 | 0.00 | 0.88 | 0.00 |
| Gibraltar Strait | 1.00 | 0.50 | 0.96 | 1.00 | 0.75 | 1.00 |
| Hormuz Strait | 0.50 | 0.25 | 0.86 | 1.00 | 0.62 | 0.50 |
| Panama Canal | 1.00 | 1.00 | 1.00 | 1.00 | 0.80 | 1.00 |
| Korean Strait | 1.00 | 0.75 | 1.00 | 1.00 | 0.22 | 1.00 |
| English Channel | 1.00 | 0.00 | 1.00 | 1.00 | 0.31 | 1.00 |
| Florida Strait | 1.00 | 0.50 | 1.00 | 1.00 | 0.84 | 1.00 |
| Dagu Strait | 1.00 | 1.00 | 1.00 | 1.00 | 0.97 | 1.00 |
| Zonggu Strait | 1.00 | 0.75 | 1.00 | 1.00 | 0.93 | 1.00 |
| Mindoro Strait | 0.00 | 1.00 | 0.98 | 1.00 | 0.88 | 0.00 |
| Windward Strait | 1.00 | 0.75 | 1.00 | 1.00 | 1.00 | 0.50 |
| Mona Strait | 1.00 | 0.75 | 0.98 | 1.00 | 1.00 | 1.00 |

**Table 5.** Standardized adaptivity index data for the key nodes.

| Index / Key Node | Alternative | Related Organization | Security Norm | Legal Policy | International Cooperation |
|---|---|---|---|---|---|
| Taiwan Strait | 0.33 | 0.33 | 1.00 | 0.60 | 0.25 |
| Malacca Strait | 1.00 | 1.00 | 1.00 | 0.80 | 1.00 |
| Mande Strait | 0.33 | 1.00 | 0.00 | 0.20 | 0.00 |
| Suez Canal | 0.33 | 0.33 | 1.00 | 1.00 | 0.00 |
| Lombok Strait | 1.00 | 0.50 | 0.00 | 0.40 | 0.00 |
| Sunta Strait | 1.00 | 0.50 | 0.00 | 0.20 | 0.00 |
| Makassar Strait | 0.67 | 0.50 | 0.00 | 0.20 | 0.00 |
| Gibraltar Strait | 0.33 | 0.33 | 1.00 | 0.40 | 0.00 |
| Hormuz Strait | 0.00 | 0.17 | 0.00 | 0.40 | 0.25 |
| Panama Canal | 0.33 | 0.33 | 1.00 | 0.00 | 0.00 |
| Korean Strait | 0.33 | 0.67 | 0.00 | 0.20 | 0.00 |
| English Channel | 0.33 | 0.50 | 0.00 | 0.20 | 0.00 |
| Florida Strait | 0.67 | 0.67 | 0.00 | 0.20 | 0.00 |
| Dagu Strait | 0.67 | 0.33 | 0.00 | 0.20 | 0.00 |
| Zonggu Strait | 0.67 | 0.33 | 0.00 | 0.20 | 0.00 |
| Mindoro Strait | 0.67 | 0.17 | 0.00 | 0.00 | 0.00 |
| Windward Strait | 0.67 | 0.00 | 0.00 | 0.00 | 0.00 |
| Mona Strait | 0.67 | 0.00 | 0.00 | 0.00 | 0.00 |

3.3.2. Establishment of the Hierarchy Structure of Key Nodes

In this paper, SPSS 20 software was used to analyse the vulnerability and adaptivity index values, and the cumulative variance contributions and factor load matrices were calculated.

1. The calculated cumulative variance contributions are shown in Tables 6 and 7.

**Table 6.** Extraction of the vulnerability index.

| Vulnerability Index | Principal Component 1 | Principal Component 2 | Principal Component 3 |
|---|---|---|---|
| Cumulative variance contribution (%) | 48.018 | 70.042 | 87.431 |

Table 6 shows that the cumulative variance contribution of the first three principal components is 87.431%, indicating that the first three principal components represent 87.431% of the information for all indices. Thus, the first three principal components can be used to represent all of the vulnerability index data.

**Table 7.** Extraction of the adaptivity index.

| Adaptivity Index | Principal Component 1 | Principal Component 2 | Principal Component 3 |
|---|---|---|---|
| Cumulative variance contribution (%) | 40.294 | 63.347 | 85.610 |

Table 7 shows that the cumulative variance contribution of the first three principal components reaches 85.610%, indicating that the first three principal components represent 85.610% of the information for all indices. Thus, the first three principal components can be used to represent all of the adaptivity index data.

2. The calculated factor load matrices are shown in Tables 8 and 9.

**Table 8.** Factor load matrix for the vulnerability index.

| Vulnerability Factors | Principal Component | | |
|---|---|---|---|
| | 1 | 2 | 3 |
| Military conflict | 0.945 | −0.044 | 0.189 |
| Coastal political stability | 0.898 | −0.221 | 0.290 |
| Pirates | 0.793 | 0.425 | −0.243 |
| Maritime terrorism | 0.723 | 0.312 | −0.022 |
| Accidents | −0.137 | 0.903 | −0.255 |
| Military bases | −0.358 | 0.556 | 0.743 |

Table 8 shows that military conflicts, coastal political stability, piracy, and maritime terrorism have high factor loads for principal component 1, and that for military conflict reaches more than 90%. Thus, these four indices are represented by principal component 1. Similarly, accidents have a high factor load for principal component 2, so principal component 2 reflects the effect of accidents. The value for military bases is high for principal component 3, and principal component 3 reflects the number of military bases.

**Table 9.** Factor load matrix for the adaptivity index.

| Adaptivity Factors | Principal Component | | |
|---|---|---|---|
| | 1 | 2 | 3 |
| Legal policy | 0.857 | −0.199 | 0.045 |
| International cooperation | 0.818 | 0.257 | 0.144 |
| Security norm | 0.744 | −0.474 | 0.234 |
| Alternatives | 0.044 | 0.847 | 0.481 |
| Related organizations | 0.545 | 0.504 | −0.643 |

As shown in Table 9, legal factors, international cooperation, and security norms have high factor loads for principal component 1 and are represented by principal component 1. Similarly, alternatives and related organizations have high factor loads for principal component 2 and principal component 3, respectively; thus, principal components 2 and 3 represent alternatives and related organizations, respectively.

Based on the above results, the hierarchical structure model shown in Figure 2 can be established. Notably, the importance of each layer gradually decreases from left to right.

### 3.3.3. Establishment of a Catastrophe Model Based on the Hierarchical Structure of Key Nodes

According to Figure 2 and Table 1, a coupled risk catastrophe model of the key nodes in Chinese maritime transport can be established, as shown in Figure 3.

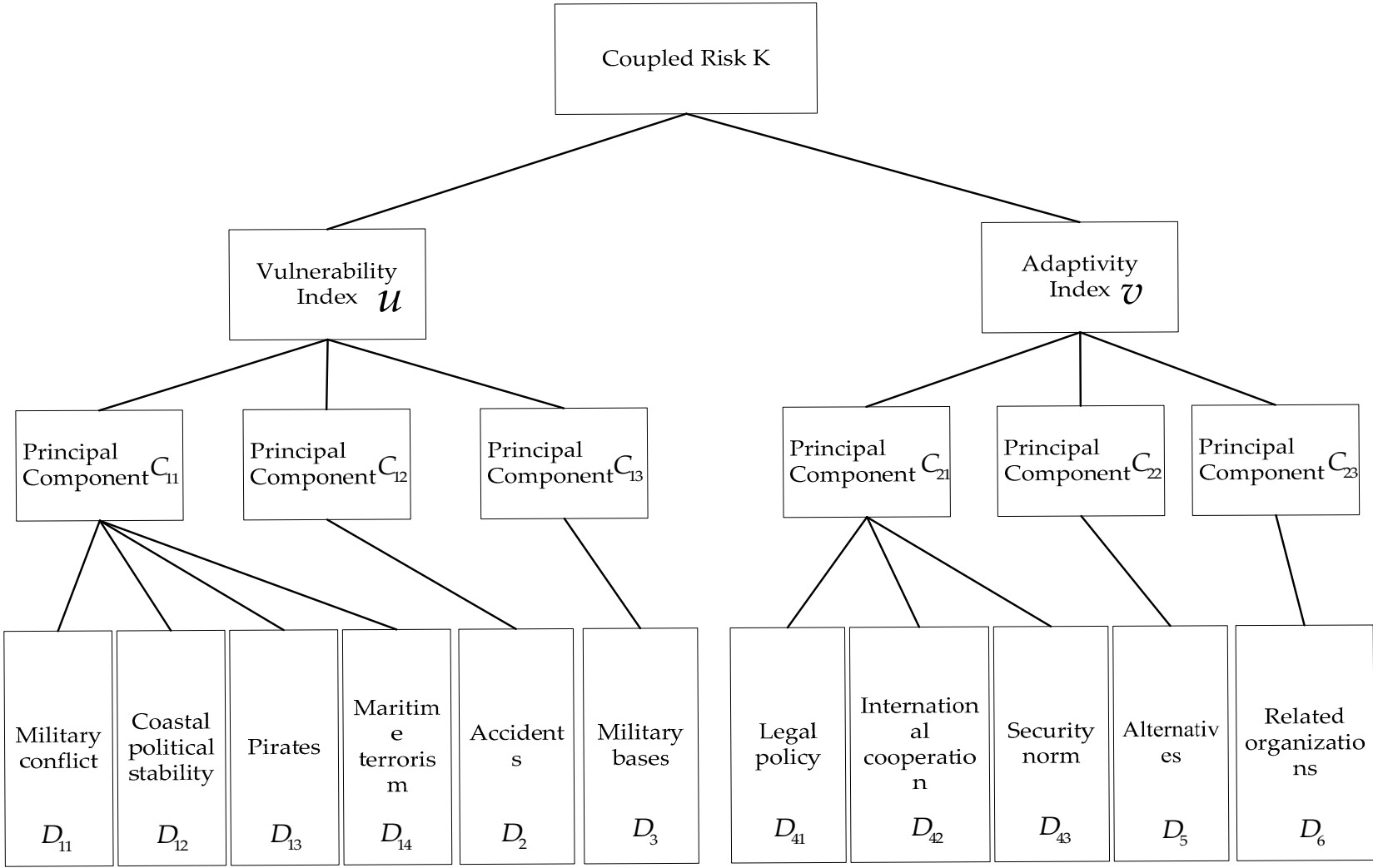

**Figure 2.** The hierarchical structure of the coupled risk of key nodes.

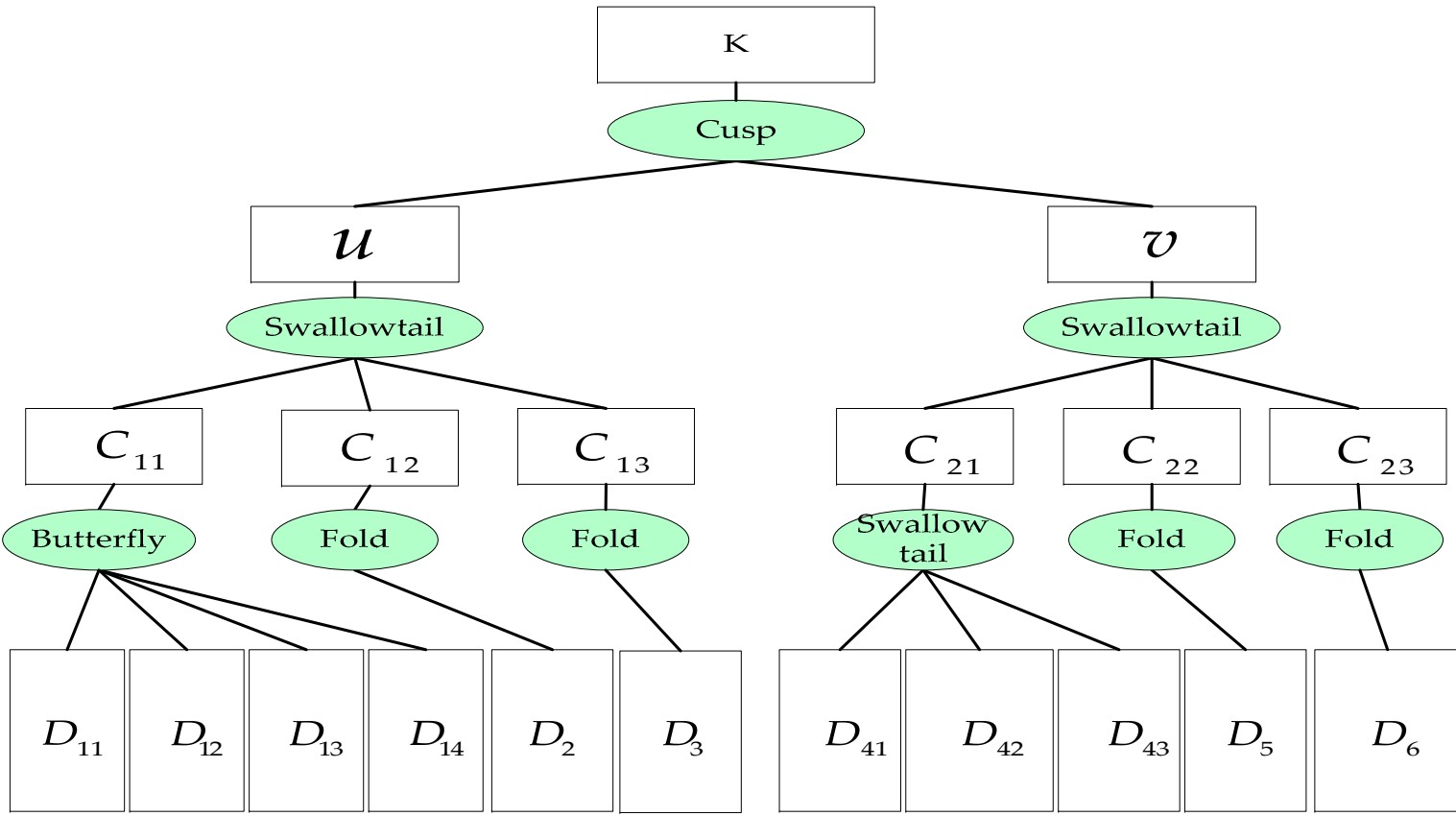

**Figure 3.** The hierarchical structure of the coupled risk of key nodes.

## 3.4. Results and Analysis

### 3.4.1. Calculation Results

According to the third part of Section 2.2.2, the catastrophe model identified in Figure 3, and the normalization formula corresponding to the catastrophe model in Table 1, the coupling degree values of key nodes in Chinese maritime transportation are divided into 5 levels, as shown in Table 10. Security level 1 is the lowest level of security, and the level of security increases to level 5.

**Table 10.** Security level of key nodes.

| Security Level | I | II | III | IV | V |
|---|---|---|---|---|---|
| Scale $k$ | [0, 0.9035) | [0.9035, 0.9436) | [0.9436, 0.9680) | [0.9680, 0.9859) | [0.9859, 1] |

1. The comprehensive coupled risk of key nodes in maritime transport is shown in Table 11.

**Table 11.** Comprehensive measurement results for key nodes.

| Key Node | Rank | Security Level | Coupling Degree Value | Adjusted Coupling Degree Value |
|---|---|---|---|---|
| Taiwan Strait | 1 | III | 0.9617 | 0.5480 |
| Malacca Strait | 14 | I | 0.8771 | 0.1941 |
| Mande Strait | 9 | II | 0.9225 | 0.2945 |
| Suez Canal | 5 | III | 0.9472 | 0.4301 |
| Lombok Strait | 10 | II | 0.9107 | 0.2356 |
| Sunta Strait | 11 | II | 0.9076 | 0.2205 |
| Makassar Strait | 12 | I | 0.8986 | 0.1989 |
| Gibraltar Strait | 2 | III | 0.9558 | 0.5004 |
| Hormuz Strait | 15 | I | 0.8661 | 0.1917 |
| Panama Canal | 3 | III | 0.9533 | 0.4795 |
| Korean Strait | 8 | II | 0.9268 | 0.3164 |
| English Channel | 16 | I | 0.8362 | 0.1851 |
| Florida Strait | 7 | III | 0.9463 | 0.4226 |
| Dagu Strait | 4 | III | 0.9500 | 0.4526 |
| Zonggu Strait | 6 | III | 0.9464 | 0.4232 |
| Mindoro Strait | 13 | I | 0.8896 | 0.1969 |
| Windward Strait | 18 | I | 0.8338 | 0.1846 |
| Mona Strait | 17 | I | 0.8360 | 0.1850 |

2. The coupling degrees of key nodes in maritime transportation in terms of the vulnerability and adaptivity indices are shown in Table 12.

**Table 12.** Coupling degree values of the vulnerability and adaptivity indices of key nodes.

| Key Node | Vulnerability Index Coupling Value ($u$) | Adaptivity Index Coupling Value ($v$) |
|---|---|---|
| Taiwan Strait | 0.9408 | 0.8665 |
| Malacca Strait | 0.5718 | 0.9941 |
| Mande Strait | 0.8847 | 0.7396 |
| Suez Canal | 0.9041 | 0.8403 |
| Lombok Strait | 0.8030 | 0.7921 |
| Sunta Strait | 0.8094 | 0.7677 |
| Makassar Strait | 0.7927 | 0.7459 |
| Gibraltar Strait | 0.9565 | 0.8140 |
| Hormuz Strait | 0.8982 | 0.4827 |
| Panama Canal | 0.9876 | 0.7606 |
| Korean Strait | 0.9141 | 0.7231 |
| English Channel | 0.6075 | 0.7119 |
| Florida Strait | 0.9628 | 0.7571 |

**Table 12.** *Cont.*

| Key Node | Vulnerability Index Coupling Value ($u$) | Adaptivity Index Coupling Value ($v$) |
|----------|------------------------------------------|----------------------------------------|
| Dagu Strait | 0.9985 | 0.7308 |
| Zonggu Strait | 0.9842 | 0.7308 |
| Mindoro Strait | 0.8953 | 0.5780 |
| Windward Strait | 0.9795 | 0.3116 |
| Mona Strait | 0.9880 | 0.3116 |

3. The results of the analysis of the security status of the key node security system for maritime transport are shown in Table 13.

**Table 13.** Security status of the security system of key nodes.

| Key Node | $v$ | $[\frac{8}{27}u^3]^{1/2}$ | Status |
|----------|-----|---------------------------|--------|
| Taiwan Strait | 0.8665 | 0.4967 | Stable |
| Malacca Strait | 0.9941 | 0.2353 | Stable |
| Mande Strait | 0.7396 | 0.4530 | Stable |
| Suez Canal | 0.8403 | 0.4679 | Stable |
| Lombok Strait | 0.7921 | 0.3917 | Stable |
| Sunta Strait | 0.7677 | 0.3963 | Stable |
| Makassar Strait | 0.7459 | 0.3842 | Stable |
| Gibraltar Strait | 0.8140 | 0.5092 | Stable |
| Hormuz Strait | 0.4827 | 0.4634 | Stable |
| Panama Canal | 0.7606 | 0.5342 | Stable |
| Korean Strait | 0.7231 | 0.4757 | Stable |
| English Channel | 0.7119 | 0.2578 | Stable |
| Florida Strait | 0.7571 | 0.5143 | Stable |
| Dagu Strait | 0.7308 | 0.5431 | Stable |
| Zonggu Strait | 0.7308 | 0.5315 | Stable |
| Mindoro Strait | 0.5780 | 0.4611 | Stable |
| Windward Strait | 0.3116 | 0.5277 | Unstable |
| Mona Strait | 0.3116 | 0.5346 | Unstable |

### 3.4.2. Analysis of the Results

Table 11 indicates that the overall security level of the key nodes in Chinese maritime transportation is relatively low, and most security levels are I, II, and III. Table 12 shows that the reason for these results is that the vulnerability index has a high coupled risk and a considerable impact on the overall security. The Taiwan Strait has the highest degree of coupling and the highest degree of security compared with the other key nodes, and the Windward Strait has the lowest coupling degree and the lowest degree of security compared with the other key nodes. The values for the Strait of Malacca, which links the important maritime transport corridors between East and West Asia, and the Strait of Hormuz, which is an important European maritime transport and strategic channel, are also relatively low. In addition, the difference in security among the key nodes with the same security level is highly variable. As shown in Figure 4, the degrees of security of the Taiwan Strait and the Florida Strait, which are both classified as level III, are very different. Table 12 shows that the coupling degree of the vulnerability index of the Florida Straits is larger than that of the Taiwan Strait, but the coupling degree of the adaptivity index is much smaller than that of the Taiwan Strait, thus causing this difference.

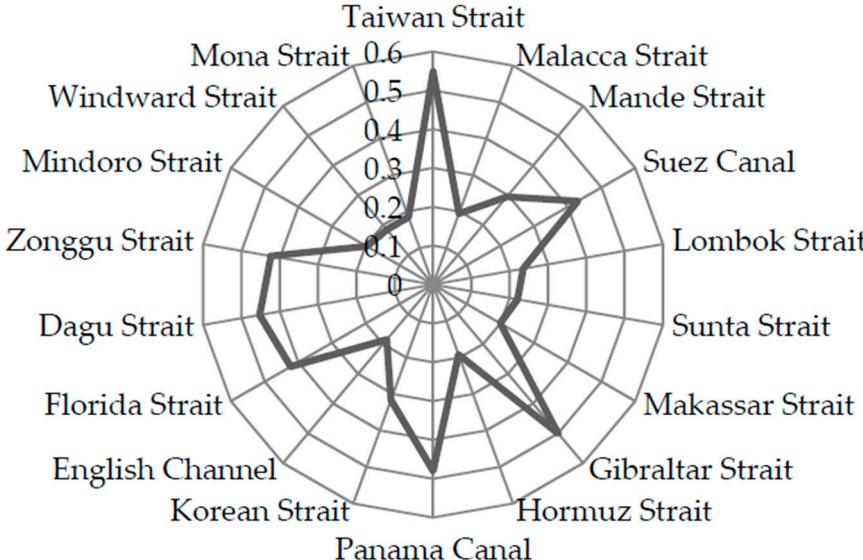

**Figure 4.** Comparison of the security levels of key nodes in maritime transport.

From the perspective of the vulnerability index, the vulnerability index value of the Strait of Malacca is the lowest, indicating that the risk factors have a considerable impact on this key node. From the perspective of the adaptivity index, the coupling degree of the adaptivity index of the Strait of Malacca is large, indicating that there are important risk prevention measures related to other key nodes but that the overall security of the Strait of Malacca is not high.

From the perspective of key nodes security system status, most of the security systems of the key nodes in Chinese maritime transport are in a stable state, as shown in Table 13. Only the Windward Strait and the Mona Strait are in an unstable state, indicating that the security systems of these nodes are prone to catastrophes. The reason for this can be seen in Table 12, which shows that the coupling values of the two strait vulnerability indicators are relatively high and the coupling values of the adaptive indicators are relatively low. Therefore, from the perspective of the whole system, the vulnerability is much higher than the adaptability, which leads to the instability of the system. Further analysis of the original statistics indicates that the military risk, accident risk, and piracy risk of the Windward Strait and the Mona Strait are higher than their corresponding defence measures. Therefore, the above results are caused and the appropriate departments should take relevant measures to improve the security level of the system. In addition, taking the Strait of Malacca as an example, although it is currently in a stable state from the perspective of security systems, the overall security level of the Straits of Malacca is low; thus, appropriate departments should also lay out relevant measures in advance.

## 4. Conclusions

In this paper, an improved catastrophe model was used to measure the coupled risk of key nodes in Chinese maritime transport. This approach provides a new concept to accurately describe and analyse the security of key nodes in maritime transport. The establishment of a hierarchical structure model based on PCA avoids subjectively establishing a measurement system, mitigates the inaccuracy of determining the importance of influential factors, and accurately describes the actual situation and problem. In addition, the calculation results are improved compared to those of the traditional method, yielding results that are conducive to in-depth analysis. Based on specific examples, a simulation is performed. The results show that the Taiwan Strait has the lowest risk and that the Windward Strait has the highest risk. The vulnerability and adaptivity indices have different degrees of impact on the security of different key nodes.

For the analysis of the key risk coupling value of the key nodes, the appropriate departments can know the current relative security level of each key node, and analysis of the adaptivity and

vulnerability indicators of the key nodes can be used to know the current preventive measures and the current threat situation, providing the basis for relevant departments to formulate the corresponding policies or preventive measures. In addition, for the analysis of the security status of key node systems, it is possible to know the stability of the system status, which provides a reference for the urgency of the relevant departments to take precautionary measures and to determine whether a ship's navigation passes through this key node. Therefore, this paper analyses the risk coupling of key nodes, providing not only reference for users and managers of key nodes, but also a basis for appropriate departments to formulate relevant policies and take security measures.

Although the model proposed in this paper has some practical significance for risk coupling assessment of maritime transport nodes, it still has some limitations. Firstly, the model proposed in this paper needs to analyse the correlation of the basic data, the inaccuracy or change of the data will directly affect the construction of the model, which will affect the accuracy of the results. Secondly, when using the principal component analysis method, the cumulative contribution rate of the first few extracted principal components should be guaranteed to reach a high level, and the ambiguity of the cumulative contribution rate calculated will affect the results of principal component selection. In addition, the data in this paper are the average data in the research period. As the data changes with the development of time, therefore, different data selection times need to be considered in different research periods. The results of this paper are relative. Whether the number of nodes is changed or the data of the node is changed, the results will change. Therefore, the nodes of the study must be clear.

**Author Contributions:** The paper was written by B.D.L. and J.L. (Jing Li) and revised and checked by J.L. (Jing Li) and J.L. (Jing Lu). All authors read and approved the final manuscript.

**Funding:** This research was funded by the Fundamental Research Funds for the Central Universities (3132019302), Humanities and Social Science Fund of the Ministry of Education of China (16YJAZH030) and the Key Project of the National Societal Science Foundation of China (18VHQ005).

**Acknowledgments:** We are thankful to anonymous reviewers and editors for their helpful comments and suggestions.

**Conflicts of Interest:** The authors declare no conflict of interest.

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
