# Peer review of "Research on the Coupled Risk of Key Nodes in Maritime Transport Based on Improved Catastrophe Theory"

_sustainability, doi:10.3390/su11174640_

Round 1

Reviewer 1 Report

This study examined the risk issues of some key maritime transportation nodes based on principal component analysis and catastrophe theory. While the study generates some interesting insights, I have some major concerns regarding the motivation, literature review and general discussion of this paper. My detailed comments are as follows:

1) Title: Abbreviations (PCA and CT) are often not recommended to be used in the article title. I suggest the authors to reconsider the article title.

2) Motivation: The research motivation is not well established in the introduction section. In particular, the authors discussed the progress of safety- and security-related research in maritime transportation, including the emerging research on risk coupling mechanisms and the adoption of catastrophe theory. However, the authors fail to show a) what are the key research gaps in the existing literature that the current study intends to address, b) how does the current research differentiate from the existing studies, and c) what are the key contributions of the current study which create new knowledge in this field. Answers to these key questions need to be reflected in the introduction section to better motivate the research.

3) Literature review: A literature review section is lacking in this study. I understand that the authors provide a brief review of extant literature in the introduction section. However, without a separate section of literature review, some important / recent studies are overlooked. I would suggest the authors to seriously consider developing a separate section for literature review. Below are some studies related to the current research:

Gao, T., Lu, J., The impacts of strait and canal blockages on the transportation costs of the Chinese fleet in the shipping network, Maritime Policy and Management

46(6), pp. 669-686

Wu, D., Wang, N., Yu, A., Wu, N., Vulnerability analysis of global container shipping liner network based on main channel disruption, Maritime Policy and Management

46(4), pp. 394-409

Cao, X., Lam, J.S.L., Simulation-based severe weather-induced container terminal economic loss estimation, Maritime Policy and Management,46(1), pp. 92-116

Liu, N., Gong, Z., Xiao, X., Disaster prevention and strategic investment for multiple ports in a region: cooperation or not, Maritime Policy and Management,45(5), pp. 585-603

Fu, S., Yan, X., Zhang, D., Zhang, M., Risk influencing factors analysis of Arctic maritime transportation systems: a Chinese perspective, Maritime Policy and Management45(4), pp. 439-455

3) Scope (key nodes in Chinese maritime transportation): In table 2, the authors list the key nodes under examination in this study. The authors claim that these nodes are “important” to Chinese maritime transportation. However, can the authors please clarify a) why these nodes are important? In what sense? Supporting statistics are needed to justify the importance, b) Are these nodes only important to Chinese maritime transportation? What is covered under “Chinese” maritime transportation? I suppose that maritime transportation should be international? C) why only straits and canals are targeted? How about some major ports and port clusters?

4) Terminology: through the manuscript, the terms “safety” and “security” seem to be used interchangeably. However, a common understanding is that there is a subtle difference between these two terms especially in risk-related research. I would suggest the authors to take caution when using these two terms. Otherwise, regulate the terminology by using only one term.

5) The authors propose vulnerability and adaptivity-based index based on some second-level index. However, I wonder what is the basis for this index system (supporting literature)? Or is it a new index system proposed only in this study? Some supporting literature would be good.

6) Conclusion: Conclusion section needs to be improved. In particular, managerial implications and theoretical contributions of this study need to be more thoroughly discussed.

Reviewer 2 Report

The manuscript presents a high-level approach to the assessment of security of navigation in certain crucial points. 

Although the concept is presented clearly with IMRAD kept, I have some major and minor comments.

Major:

Risk coupling is not defined properly, readers unfamiliar with the concept may find it difficult to understand the paper; 'Safety' and 'security' terms are used as if they were same. They are not - security refers to intentional and unwelcome acts of humans while safety refers to more random, unintentional failures of the system. It must be corrected and only security shall be referred to, or the scope of the paper must be changed; The paper is too sino-centric. It presents maritime transportation nodes' significance to the security of Chinese economy. Meanwhile, seaborne transportation is a global industry and it does not really matter what perspective is applied. The application of Chinese perspective must either be supported by proper references or removed; Catastrophe theory is also not defined or introduced properly, as well as Principal Components; The significance of the findings for either academic community or wider audience is not clearly presented; The results seem unreliable as they indicate Strait of Hormuz to be secure for shipping, in view of recent events.

Minor:

Line 32: there is no 'Turkish Strait', but 'Turkish Straits';

Line 100: please expand this paragraph and include practical considerations;

Line 108: please explain the link between social sciences and application of the theory in maritime security study;

Line 186: why is the link between degree of coupling and risk level indicated to exist only 'in this paper'. Isn't it a general relationship?

Line 228: this sentence is unclear.

Line 233: please provide reference for 'many countries taking measures...';

Table 3: how were particular indices/factors evaluated? Where does the data come from? 

Line 251: please provide exact references of the data sources;

Table 4 and 5: please rearrange indices to match their order in Table 3 for better understanding;

Table 10: where do the values come from? How were they obtained?

Line 335: how would you comment the fact that Mona and Windward Strait are not safe while Malacca Strait is safe? This seems against common understanding of maritime security and requires explanation or comment;

Line 351: please discuss the significance of the study in more detail.

Taking the above into consideration, I believe that the manuscript contains interesting approach, but must be improved.

Round 2

Reviewer 1 Report

I would like to thank the authors for addressing my comments. After the previous round revisions, I still have some minor concerns which are listed below: 

The introduction is now separated into three sections. However, the three sections are not well linked with each other and the overall discussion is not coherent after the revision. I would suggest the authors to add a starting paragraph at the beginning that provides an overview of the introduction section. Also linkage paragraphs are necessary to link the three separate sections.  In response to my comment 3 (last round), the authors cited all the references I provided. However, I don't think these references are well integrated into existing discussion. Please revise the relevant sections. Also please do not feel that it is my "request" to cite any of the papers.  I am not convinced by the authors response to my comment 4. I would suggest the authors to add an additional column in Table 2 to explain the importance to China of the selected straits and channels. 

Reviewer 2 Report

The paper has improved greatly. 

However, I am still missing two aspects:

Discussion of potential uncertainties related to the data, algorithms, equations etc. as well as consequences of data inaccuracy, for instance; A clear statement on applicability of the results of the study would also be beneficial. Who can use the results and how?

Once these are completed, I can consider the paper publication-ready.
